# MXene/Gelatin/Polyacrylamide Nanocomposite Double Network Hydrogel with Improved Mechanical and Photothermal Properties

**DOI:** 10.3390/polym14235247

**Published:** 2022-12-01

**Authors:** Zeyu Zhang, Yang Hu, Huiling Ma, Yicheng Wang, Shouchao Zhong, Lang Sheng, Xiang Li, Jing Peng, Jiuqiang Li, Maolin Zhai

**Affiliations:** 1Beijing National Laboratory for Molecular Sciences, Radiochemistry and Radiation Chemistry Key Laboratory of Fundamental Science, The Key Laboratory of Polymer Chemistry and Physics of the Ministry of Education, College of Chemistry and Molecular Engineering, Peking University, Beijing 100871, China; 2School of Materials Design and Engineering, Beijing Institute of Fashion Technology, Beijing 100029, China

**Keywords:** radiation synthesis, nanocomposite double-network hydrogel, mechanical properties, photothermal properties

## Abstract

The development of smart hydrogel with excellent mechanical properties and photothermal conversion capability is helpful in expending its application fields. Herein, a MXene/gelatin/polyacrylamide (M/G/PAM) nanocomposite double network (NDN) hydrogel was synthesized by γ-ray radiation technology for the first time. Compared with gelatin/polyacrylamide double network hydrogel, the optimized resultant M_3_/G/PAM NDN hydrogel shows better mechanical properties (tensile strength of 634 ± 10 kPa, compressive strength of 3.44 ± 0.12 MPa at a compression ratio of 90%). The M_3_/G/PAM NDN hydrogel exhibits a faster heating rate of 30 °C min^−1^, stable photothermal ability, and mechanical properties even after 20 cycles of on–off 808 nm near-infrared (NIR) laser irradiation (1.0 W cm^−2^). Furthermore, the temperature of M_3_/G/PAM NDN hydrogel can be increased rapidly from 25 °C to 90 °C in 10 s and could reach 145 °C in 120 s under irradiation by focused NIR laser irradiation (56.6 W cm^−2^). The high mechanical property and photothermal properties of M/G/PAM hydrogel are ascribed to the formation of double network and uniform hydrogen bonding between MXene and gelatin and PAM polymers. This work paves the way for construction of photothermal hydrogels with excellent mechanical properties.

## 1. Introduction

Hydrogel, a three-dimensional network polymer material with high water content [1], has good material compatibility, which can be compounded with other materials to endow hydrogel with different functions for the applications in various fields [2,3]. However, with the long-term use of traditional single-component hydrogel, its inherent weak mechanical properties have emerged as a challenge to be solved. At present, more and more hydrogels with high mechanical properties have been reported, for instance, double network (DN) hydrogels [4], nanocomposite hydrogels [5,6], ionically cross-linked hydrogels [7], and sliding-ring hydrogels [8], etc. Gong et al. reported the first DN hydrogel, consisting of poly(2-acrylamido-2-methylpropanesulfonic acid) and polyacrylamide (PAM), which overcomes the weak mechanical properties of the traditional single-component hydrogel [9]. Since then, other DN hydrogels based on chemical bonds or physical interactions have been widely reported [10,11,12,13].

Gelatin, as a biodegradable protein, is obtained from collagen, which has wide applications in biomedical, pharmaceutical, and food materials [14,15]. Gelatin can be formed by a physical crosslinking network (triple helixes) during the process of heating–cooling. However, its poor mechanical properties greatly limit its applications. Blending the gelatin with other polymers to synthesize DN hydrogel is an effective method to address this problem. For example, Fan et al. synthesized shape memory PAM/gelatin hydrogels with adjustable mechanical strength via a facile one-pot method and found the PAM/gelatin hydrogel containing 5% *w*/*v* of gelatin exhibited the optimum mechanical properties (0.57 MPa tensile strength, 1556% elongation at break, 0.06 MPa compression strength and 0.38 MPa Young’s modulus) [16]. Apparently, the mechanical property was significantly improved by fabricating the DN structure, but the mechanical property still remains to be improved on the whole.

According to previous reports, the interaction between highly hydrophilic groups (hydroxyl, amino, etc.), of the hydrogel and other nanomaterials can further improve the mechanical properties of hydrogels [17]. Ti_3_C_2_T*_x_* exhibits excellent hydrophilicity, light absorption capacity, photothermal conversion efficiency, and mechanical properties [18], which can be compounded with hydrogel to enhance mechanical properties [19]. Li et al. synthesized a nanocomposite double-network (NDN) hydrogel composed of polyacrylamide-co-acrylic acid/chitosan covalent network reinforced by Ti_3_C_2_T*_x_* MXene nanosheets with excellent mechanical properties [20]. The uniformly distributed Ti_3_C_2_T*_x_* within the hydrogel not only greatly improves the mechanical properties due to the formation of hydrogen bonds with the hydrogel matrix [21], but also further endows its photothermal ability (temperature rose from 20 °C to 50 °C under irradiation 1.0 W cm^−2^ for about 5 min) [22]. Yang et al. prepared a novel Ti_3_C_2_T*_x_*/poly(N-isopropylacrylamide) composite hydrogel with excellent photothermal properties under near-infrared (NIR) laser irradiation (the temperature of the water with the incorporated composite hydrogels rose by over 20 °C under 0.8 W focused irradiation in 5 min) [23]. However, the mechanical performance of these composite hydrogels should be further improved. It is expected that the fabrication of DN hydrogel will improve the mechanical performance of the MXene-based hydrogel, and the preparation of MXene NDN hydrogels has rarely been reported.

Recently, we synthesized the agarose/polyacrylamide DN hydrogel, copper-alginate/polyacrylamide DN hydrogel, and agarose/Ti_3_C_2_T*_x_*-cosslinked-polyacrylamide DN hydrogel with extremely high mechanical properties by using the γ-ray radiation-induced method [24,25,26]. It was found that the γ-ray radiation technology has great advantages in synthesizing high-performance DN hydrogel. Herein, a MXene/gelatin/polyacrylamide (M/G/PAM) NDN hydrogel with excellent mechanical and photothermal properties was synthesized by the γ-ray radiation technology. The introduced MXene can not only increase the network density of the hydrogel to improve the mechanical properties but also further endow the hydrogel with excellent photothermal ability. The above results were further confirmed by using different characterization. The mechanical properties of hydrogels were investigated by tensile and compression tests. At last, the influences of different contents of MXene within the M/G/PAM NDN hydrogel on the photothermal conversion performance were explored under 808 nm NIR laser irradiation with different power densities. The photothermal cyclic stability of optimal M_3_/G/PAM NDN hydrogel and its mechanical properties in the repeatedly on–off NIR irradiation (1.0 W cm^−2^) were also tested.

## 2. Materials and Methods

### 2.1. Materials

Ti_3_AlC_2_ (99%) was purchased from Jilin 11 Technology Co., Ltd. with particle size of less than 19 μm. LiF was obtained from Shanghai Aladdin Biochemical Technology Co., Ltd. Hydrochloric acid (HCl, 12 mol L^−1^) solution was obtained from Beijing Chemical Works. Acrylamide (AM, 99%) and *N,N’*-methylene-bis-acrylamide (MBA, 99%) was supplied from Beijing Bailingwei Technology Co., Ltd. Gelatin was of chemically pure and was obtained from Sinopharm Chemical Reagent Co., Ltd. All the chemicals were used without further purification.

### 2.2. Synthesis of Few-Layered Ti_3_C_2_T_x_ MXene

An amount of 8.0 g of LiF was added to 100.0 mL HCl (12 mol L^−1^) aqueous solution to obtain a uniformly dispersed solution, then 3.0 g Ti_3_AlC_2_ was slowly added under stirring. After reacting for 72 h at 40 °C, the solution was repeatedly centrifuged at 3500 rpm to separate a precipitate and washed with deionized water to a constant pH, obtaining the multi-layered Ti_3_C_2_T*_x_* sediment. The above multi-layered Ti_3_C_2_T*_x_* sediment was dispersed in 150 mL deionized water and ultrasonicated for 1 h. Finally, the product of ultrasonic was centrifuged at 3000 rpm for 30 min to separate the supernatant, obtaining the few-layered Ti_3_C_2_T*_x_* aqueous solution. Then, it was transferred to a sealed bottle and stored at 4 °C under N_2_ atmosphere. The yield and solution concentration of few-layered Ti_3_C_2_T*_x_* were calculated.

### 2.3. Preparation of M/G/PAM NDN Hydrogel

Firstly, gelatin was dissolved in pure water with agitation at 45 °C for 1 h to form a homogeneous and transparent solution. Then, AM (monomer) and few-layered Ti_3_C_2_T*_x_* suspension (semiconductor materials), MBA (cross-linking agent, 10 mg mL^−1^, MBA/AM molar ratio of 0.00750) were added to the above solution under constant stirring at 45 °C. Finally, the resultant mixture solution was poured into glass mold and was stored for 1 h in a refrigerator with temperature of 4 °C, and then irradiated with ^60^Co source from Department of Applied Chemistry of Peking University.

Based on our previous work [24], the optimum dose and dose rate was fixed at 300 Gy and 10 Gy min^−1^. The prepared hydrogels containing 0.035 wt%, 0.070 wt%, 0.105 wt%, and 0.140 wt% few-layered Ti_3_C_2_T*_x_* were named as M_1_/G/PAM NDN hydrogel, M_2_/G/PAM NDN hydrogel, M_3_/G/PAM NDN hydrogel and M_4_/G/PAM NDN hydrogel, respectively. For comparison, gelatin single-network (G SN) hydrogel, polyacrylamide single-network (PAM SN) hydrogel, and G/PAM DN hydrogel were also prepared. Table 1 shows the proportioning parameters of as-prepared samples.

### 2.4. Characterizations

The all characterizations were performed using the freeze-dried samples. The field-emission scanning electron microscopy (SEM) was observed using a Hitachi S-4800 microscope with an acceleration voltage of 2 kV. Samples were prepared by adhering dried hydrogel to the conductive adhesive with spraying gold before observation. The X-ray diffraction (XRD) patterns of the prepared samples were measured through a Philips X’Pert Pro diffractometer equipped with an X’celerator detector using Cu-Ka radiation (λ = 1.54178 Å) at a generator voltage of 40 kV and a generator current of 40 mA from 5~85°. The Fourier transform infrared (FT-IR) spectra of the samples were determined using a Magna-IR 750 in the range of 4000–600 cm^−1^. The thermogravimetric analysis (TGA) and differential thermal analysis (DTA) were analyzed by using a Q600 SDT instrument (TA Instruments, Milford, MA, USA) from 25 to 700 °C in a heating rate of 10 °C min^−1^ with nitrogen flow rate was 100 mL min^−1^.

### 2.5. Gel Fraction Measurement

The gel fraction of samples was obtained by weighing the dry gel before and after removing the sol part. In this process, the same shape hydrogel was vacuum-dried (60 °C) until the weight is constant (*w_0_*). The samples were then immersed in deionized water for 7 days and dried again to a constant weight (*w_g_*). The gel fraction (*G_f_*) was calculated by formula (1):(1)$$G_{f}\left(\%\right)=w_{g}/w_{0\;\;}\times 100\;$$

### 2.6. Mechanical Tests

Compressive tests of hydrogels were carried out using a tensile tester Instron 3365 with compressive rates of 1 mm min^−1^. Here, hydrogel was cut into cylindrical shapes with a diameter of 14 mm and a height of 7 mm. For the cyclic compressive test, the hydrogels reached different maximum compression strains (*ε_max_*) and subsequently returned to the initial position. Compression strain (*ε*) and compression stress (σ’) are calculated by formulas (2) and (3), respectively.
(2)$$\varepsilon =\frac{\Delta h}{h_{0}}\times 100\%$$
(3)$$\sigma^{\prime }=\frac{F^{\prime }}{A_{0}^{\prime }}$$
where Δh is the compressive deformation, $h_{0}$ is the initial height, $F^{\prime }$ is the load and $A_{0}^{\prime }$ is the initial cross-section area of the hydrogel.

The tensile tests of hydrogels were carried out using the same universal testing machine with the tensile rates of 100 mm min^−1^. Before measurement, the hydrogel was cut into dumbbell shapes (gauge length of 20 mm, width of 4 mm and thickness of 1.5 mm) by using CP-25 punching machine and corresponding die. For the cyclic tensile test, the hydrogel reached different maximum strains (*λ_max_*) and subsequently returned to the initial position. Tensile strain (*λ*), tensile stress (*σ*) and energy dissipation (*T*) are calculated by Formulas (4)–(6), respectively.
(4)$$\lambda =\frac{\Delta l}{l_{0}}\times 100\%$$
(5)$$\sigma =\frac{F}{A_{0}}$$
(6)$$T=\int_{0}^{\lambda_{max}}\sigma d\lambda \;$$
where Δl is the tensile deformation, $l_{0}$ is the initial length, *F* is the load, and $A_{0}$ is the initial cross-section area of the hydrogel.

The mechanical stability of the M_3_/G/PAM NDN hydrogel was evaluated by the measurement of the tensile strength after repeated laser on–off of 808 nm NIR laser irradiation (1.0 W cm^−2^) (5 cycles, 10 cycles, 15 cycles, and 20 cycles) with some silicon oil on the surface of hydrogel to avoid the evaporation of water.

### 2.7. Photothermal Conversion Test

In order to evaluate the photothermal conversion performance, the cylindrical shape hydrogel with a diameter of 15 mm and a height of 1.5 mm was irradiated with an 808 nm NIR light laser (1.0, 1.5, 2.0, and 56.6 W cm^−2^). The distance between laser and samples is 10 cm. An infrared camera (Testo 869) was used to measure the real-time temperatures of hydrogel at the same time. The photothermal stability of the M_3_/G/PAM NDN hydrogel was evaluated by the measurement of the real-time temperature of hydrogel under the twenty successive cycles of on–off 808 nm NIR laser irradiation (1.0 W cm^−2^ and 56.6 W cm^−2^).

## 3. Results

### 3.1. Preparation and Mechanical Properties of M/G/PAM NDN Hydrogel

Figure 1 shows the synthesis process of M/G/PAM NDN hydrogel. Gelatin, as a kind of water-soluble renewable material, presented irregular conformation in the form of linear polymer chains when gelatin particles are dissolved in water at 45 °C because the temperature of 45 °C is higher than the melting point of gelatin (30 °C) [27]. However, the triple helixes and cross-link points of gelatin are formed via hydrogen bonding interaction when the temperature decreases to 4 °C [28]. Thus, the physical crosslinking network of gelatin was formed with the increase in cross-link density during the process of heating–cooling. Subsequently, the free radicals produced by γ-ray radiation induced polymerization and the crosslinking reaction of the AM monomer in the presence of crosslinking agent MBA, which interspersed with the gelatin network to form a dual network structure. At the same time, a few-layered Ti_3_C_2_T*_x_* was uniformly dispersed within the hydrogel to form hydrogen bonds with the groups in the polymer chains of gelatin and PAM, which further builds a dense crosslinked network. Compared with PAM SN and G/PAM DN, the microstructure of M/G/PAM NDN hydrogel is more complex. The gel fraction of hydrogels were further measured. It is found that the gel fraction is 87.5 ± 0.9%, 86.7 ± 0.4%, 87.8 ± 1.4%, 88.2 ± 1.4%, and 87.1 ± 1.6% with the increasing MXene content from 0 wt% to 0.140 wt%. Apparently, the gel fraction has no obvious change after adding MXene with different contents, indicating that MXene has no influence on the formation of the PAM crosslinking network.

The compression and tensile tests were carried out to investigate the mechanical properties of M/G/PAM NDN hydrogels. The compressive stress–strain curves and corresponding compressive strength of M/G/PAM NDN hydrogels with different Ti_3_C_2_T*_x_* contents are shown in Figure 2a,b, respectively. The compressive strength of M/G/PAM NDN hydrogel increases over the equipment pressure range. The compression property is investigated by comparing the compressive strength at a 90% compression rate. Notably, with the increase in Ti_3_C_2_T*_x_* content from 0 to 0.105 wt%, the compressive strength increased from 2.68 ± 0.130 MPa to 3.44 ± 0.12 MPa, because of the increase in the hydrogen bonds. Nevertheless, when the Ti_3_C_2_T*_x_* content increases to 0.140 wt%, the compressive strength slightly decreased to 3.10 ± 0.12 MPa, which is attributed to the destruction of hydrogen bonds caused by the aggregation of Ti_3_C_2_T*_x_* [26]. The tensile stress–strain curves and corresponding tensile strength of hydrogels with different Ti_3_C_2_T*_x_* contents are shown in Figure 2c,d, respectively. The results show that the tensile strength of hydrogels increases first and then decreases with the increase in Ti_3_C_2_T*_x_* content. With the increase in Ti_3_C_2_T*_x_* content from 0 to 0.105 wt%, the tensile strength increased almost three times (from 204 ± 12 kPa to 634 ± 10 kPa). The addition of Ti_3_C_2_T*_x_* as physical crosslinkers can improve the mechanical properties of G/PAM DN hydrogel, which is attributed to the formation of hydrogen bonds between Ti_3_C_2_T*_x_* and polymer chains. Additionally, the hydrogen bonds are beneficial to the formation of a uniform hydrogel network, which promotes energy dissipation during deformation. However, it can be found that when Ti_3_C_2_T*_x_* content was 0.140 wt%, the tensile strength of M_4_/G/PAM NDN hydrogel decreased to 337 ± 22 kPa. It is mainly due to the destruction of hydrogen bonds caused by the aggregation of Ti_3_C_2_T*_x_*. The influence of Ti_3_C_2_T*_x_* content on energy dissipation and Young’s modulus of hydrogel was further investigated. As shown in Figure 2e,f, the changing trend of energy dissipation and Young’s modulus is consistent with the change in tensile strength. It can be observed that the energy dissipation and Young’s modulus of hydrogel increase with the increase in Ti_3_C_2_T*_x_* content from 0 to 0.105 wt% at first. Subsequently, with the further increase in Ti_3_C_2_T*_x_* content to 0.140 wt% within the hydrogel, the energy dissipation and Young’s modulus begin to decrease. Table 2 compares the mechanical properties of G/PAM DN hydrogel and M_3_/G/PAM NDN hydrogel. M_3_/G/PAM NDN hydrogel has better tensile strength, which is almost three times that of G/PAM DN hydrogel. Therefore, the M_3_/G/PAM NDN hydrogel was selected for further investigations.

### 3.2. Structural Characterization

The structure and micromorphology of samples are measured by SEM (Figure 3). As shown in Figure 3a,b, G SN gel has an uneven pore structure, while in Figure 3c,d, the structure of PAM SN gel is regular with a pore size of 3 μm. It can be observed from Figure 3e,f that the pore size of G/PAM DN hydrogel is uneven, which is due to the interpenetrating of the gelatin network and PAM network with each other. In Figure 3g,h, the M_3_/G/PAM NDN hydrogel exhibits a regular network structure with a pore size of 5 μm, and no aggregated Ti_3_C_2_T*_x_* lamellae are observed, which proves that Ti_3_C_2_T*_x_* is uniformly dispersed within the hydrogel. By further comparing with the G/PAM DN gel, the hydrogen bonds interaction between Ti_3_C_2_T*_x_* and polymer chain within the M_3_/G/PAM hydrogel promotes the formation of uniform network structure, which is beneficial to the energy dissipation during tensile and compression.

Furthermore, the XRD characterization was used to monitor the structure of M_3_/G/PAM NDN hydrogel. Figure 4a shows the XRD spectra of the few-layered Ti_3_C_2_T*_x_*, G/PAM DN gel, and M_3_/G/PAM NDN gel. The characteristic peak of Ti_3_C_2_T*_x_* (2*θ* = 6°) does not appear in M_3_/G/PAM NDN gel, which is ascribed to the uniform dispersion of Ti_3_C_2_T*_x_* in the gel. The structure of M_3_/G/PAM NDN hydrogel was further determined by FT-IR spectra. Figure 4b shows the FT-IR spectra of the G SN gel, PAM SN gel, G/PAM DN gel, and M_3_/G/PAM NDN gel. For G SN gel, the characteristic peak of C-H bending vibrations (N-CO-CH_2_) appeared at 1361 cm^−1^ [29]. For PAM SN gel, the characteristic peaks at around 1650 cm^−1^, 3200 cm^−1^, and 3349 cm^−1^ originated from the stretching vibrations of C-O and N-H, respectively [24,30]. For G/PAM DN gel, the C=O characteristic peak slightly decreases, suggesting the formation of hydrogen bonds [16]. In addition, the characteristic peak of M_3_/G/PAM NDN gel at 1650 cm^−1^ was attributed to the stretching vibrations of C=O groups, and the peak increase with MXene incorporated into hydrogel [2]. Figure 4c,d shows the TGA and DTA curves of the above samples. For G SN gel, the weight loss between 230 °C and 430 °C is due to its thermal decomposition. The PAM SN gel shows two pyrolysis stages. The first pyrolysis stage at 200~320 °C originated from the reaction of adjacent amino groups within the PAM structure, and the second pyrolysis stage occurring at 320~450 °C is caused by the thermal degradation of the PAM skeleton. In addition, the total thermal weight loss rate is 88%. For G/PAM DN gel, two clear pyrolysis stages also appear in the temperature range of 150~250 °C and 250~450 °C, respectively. Additionally, the peak of M_3_/G/PAM NDN hydrogels at 160~170 °C originates from the weight loss of the water attached and bound by the gel [31]. In addition, the M_3_/G/PAM NDN gel exhibits a lower thermal weight loss rate of 79%. It is mainly caused by the hydrogen bond interaction between MXene and G/PAM in M_3_/G/PAM NDN gel, which could increase the carbonization yield of the gel during thermal degradation, a similar phenomenon has been observed in graphene oxide/cellulose composites [32].

### 3.3. Energy Dissipation Mechanism and Mechanical Properties of M_3_/G/PAM NDN Hydrogel

The energy dissipation ability of M_3_/G/PAM NDN hydrogel was investigated by the cyclic tensile and cyclic compression tests. According to previous literature reports [24,25], the area of the hysteresis loop is usually used to evaluate the energy dissipation ability. As can be seen from Figure 5a,b, the area of the first hysteresis loop is most obvious under different maximum strains. Apparently, the area of other hysteresis loops is not obvious, due to the DN structure change being irreversible in a short time. During the first tensile cycle, the energy dissipation mainly originates from the hydrogen bond between Ti_3_C_2_T*_x_* and the polymer chains. In the subsequent tensile cycles, the energy dissipation is provided only by polymer chains. Figure 5c,d shows the successive five-run compression tests of M_3_/G/PAM NDN hydrogel with 50% and 80% compression strain. Obviously, the cyclic compression energy dissipation of M_3_/G/PAM NDN hydrogel has the same trend as the cyclic tensile. Table 3 compares the mechanical properties of M_3_/G/PAM NDN hydrogel with other MXene-based nanocomposite hydrogels reported in the literature and illustrates M_3_/G/PAM NDN hydrogel has higher tensile strength with a lower MXene content.

### 3.4. Photothermal Conversion Performance of Hydrogels

The photothermal conversion performance of hydrogel containing 0 wt%, 0.035 wt%, 0.070 wt%, 0.105 wt%, and 0.140 wt% few-layered Ti_3_C_2_T*_x_* was measured with irradiation by 808 nm NIR laser irradiation were first investigated (1.0 W cm^−2^). Figure 6a shows the photothermal ability of the samples. With the increase in NIR irradiation time, the temperature of G/PAM DN hydrogel is nearly equal, which suggests that the G/PAM DN hydrogel has no photothermal ability. Whereas the M_3_/G/PAM NDN hydrogel exhibits a faster heating rate (30 °C min^−1^), the temperature increased from 25 °C to 45 °C in 40 s, which is better than that of other photothermal hydrogels [37,38,39]. In addition, the photothermal ability is deeply affected by power density, and the above result confirmed that the M_3_/G/PAM NDN hydrogel has the optimal photothermal ability. Therefore, the photothermal ability of M_3_/G/PAM NDN hydrogel with different power densities was tested (1.0 W cm^−2^, 1.5 W cm^−2^, and 2.0 W cm^−2^). The temperature change in M_3_/G/PAM NDN hydrogel is shown in Figure 6b. As laser power density enhances, the temperature of M_3_/G/PAM NDN hydrogel rises to 45 °C (1.0 W cm^−2^), 50 °C (1.5 W cm^−2^), and 55 °C (2.0 W cm^−2^) after 40 s of 808 nm NIR irradiation. In addition, the corresponding real-time infrared thermal images of M_3_/G/PAM NDN hydrogel were also recorded (Figure 6c). The M_3_/G/PAM NDN hydrogel still has excellent photothermal stability even after 20 cycles of on–off NIR irradiation with 1.0 W cm^−2^ power density (Figure 6d). Figure 6e shows the mechanical properties of M_3_/G/PAM NDN hydrogel upon repeated laser on–off of 808 nm NIR laser irradiation. The tensile strength remained up to 514 ± 37 kPa after 20 cycles of on–off NIR irradiation, indicating that the M_3_/G/PAM NDN hydrogel can maintain good mechanical properties after repeated NIR light irradiation and cooling. These results demonstrate that the M_3_/G/PAM NDN hydrogel displays an outstanding photothermal conversion performance and mechanical properties and can be used in many biomedical applications.

The photothermal ability of the G/PAM DN hydrogel and M/G/PAM NDN hydrogels was further measured by focused NIR laser irradiation (56.6 W cm^−2^). As shown in Figure 7a, the M/G/PAM NDN hydrogel exhibits excellent photothermal ability compared with G/PAM DN hydrogel. It can be found that the MXene fillers played a significant role as the photothermal conversion reagent within the M/G/PAM NDN hydrogel, because of its excellent light absorption, high photothermal conversion efficiency, high thermal conductivity, and large specific surface area. Figure 7b further compares the temperature change in the hydrogels with different contents of Ti_3_C_2_T*_x_* after 120 s NIR laser irradiation. Apparently, with increasing the Ti_3_C_2_T*_x_* content from 0 to 0.140 wt%, the photothermal ability of the M/G/PAM NDN hydrogels increases first and then decreases. Among them, M_3_/G/PAM NDN hydrogel has superior photothermal ability. It is proven that the content of Ti_3_C_2_T*_x_* plays a significant role in improving the photothermal conversion performance of hydrogel [36]. Compared with the M_3_/G/PAM NDN hydrogel, the M_4_/G/PAM NDN hydrogel shows weaker photothermal ability due to the long distance between MXene and heat conduction caused by aggregation of Ti_3_C_2_T*_x_*, which is consistent with the mechanical properties. At the same time, the real-time infrared thermal images of different hydrogels in different NIR laser irradiation times (0 s, 10 s, 60 s, 120s) were also recorded (Figure 7c). In addition, the photothermal cyclic stability of M_3_/G/PAM NDN hydrogel is important in practical applications. The results are shown in Figure 7d, after 20 cycles of on–off NIR laser irradiation (every cycle of NIR irradiation for 120 s), the M_3_/G/PAM NDN hydrogel could still be heated up to 143 °C, which indicates that M_3_/G/PAM NDN hydrogel has excellent photothermal stability and photothermal conversion performance.

## 4. Conclusions

In summary, MXene-embedded gelatin/PAM nanocomposite hydrogels were successfully synthesized by γ-ray radiation technology. The mechanical properties and photothermal conversion performance of the hydrogel have obviously been influenced by the content of the Ti_3_C_2_T*_x_*. The mechanical properties of the M_3_/G/PAM NDN hydrogel were enhanced by the addition of Ti_3_C_2_T*_x_* as physical crosslinkers, which is attributed to the formation of hydrogen bonds between Ti_3_C_2_T*_x_* and polymer chains. In addition, the hydrogen bonds are beneficial to the formation of a uniform hydrogel network, which promotes energy dissipation during deformation. Furthermore, M_3_/G/PAM NDN hydrogel exhibits excellent photothermal ability, photothermal stability, and faster heating rate under the irradiation of an 808 nm NIR laser. More important, the M_3_/G/PAM NDN hydrogel has good mechanical properties after repeated NIR light irradiation at 1.0 W cm^−2^. These results indicate the M_3_/G/PAM NDN hydrogels have great application potential in the field of the smart hydrogel.

## Figures and Tables

**Figure 1 polymers-14-05247-f001:**
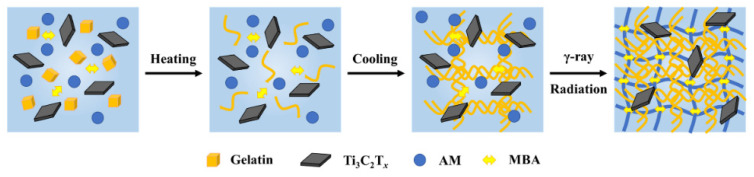
Schematic illustration of the synthesis and network structure of M/G/PAM NDN hydrogel.

**Figure 2 polymers-14-05247-f002:**
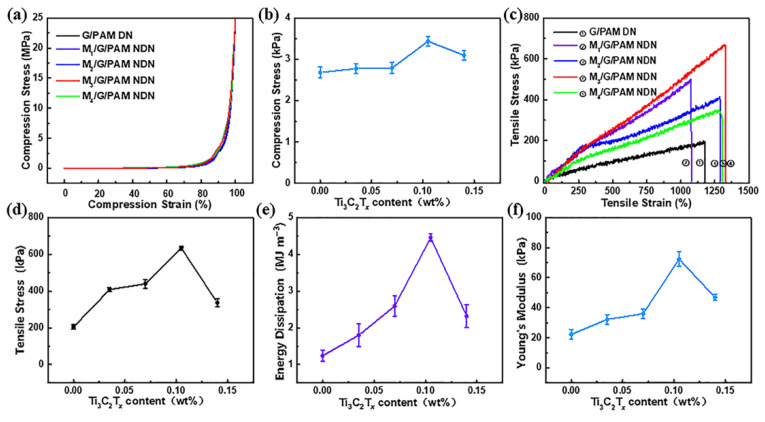
Comparison of (**a**) compressive stress–strain curves, (**b**) compressive strength (90% compression rate), (**c**) tensile stress–strain curves, (**d**) tensile strength, (**e**) energy dissipation and (**f**) Young’s modulus with different contents of Ti_3_C_2_T*_x_*.

**Figure 3 polymers-14-05247-f003:**
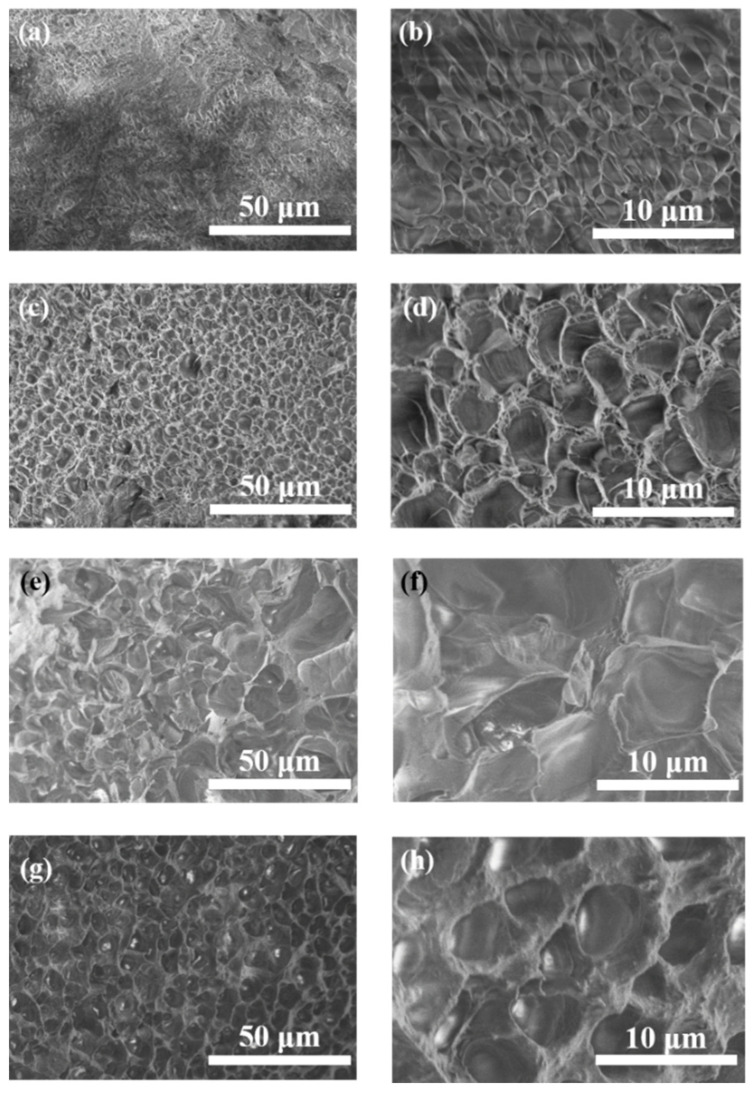
SEM images of (**a**,**b**) G SN gel, (**c**,**d**) PAM SN gel, (**e**,**f**) G/PAM DN gel and (**g**,**h**) M_3_/G/PAM NDN gel.

**Figure 4 polymers-14-05247-f004:**
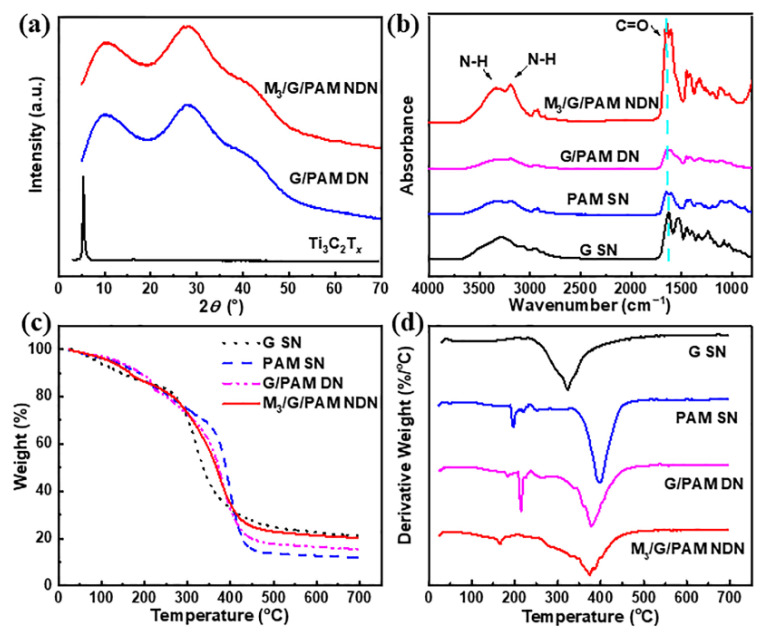
(**a**) XRD spectra of few-layered Ti_3_C_2_T*_x_*, G/PAM DN gel, and M_3_/G/PAM NDN gel; (**b**) FT-IR spectra and (**c**) TGA curves and (**d**) DTG curves of G SN gel, PAM SN gel, G/PAM DN gel, and M_3_/G/PAM NDN gel.

**Figure 5 polymers-14-05247-f005:**
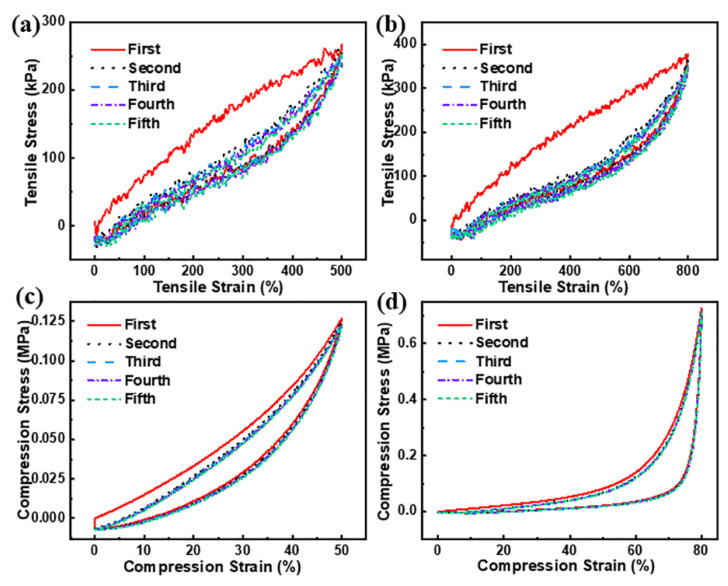
Cyclic tensile loading–unloading curves of M_3_/G/PAM NDN hydrogel at strain of (**a**) 500%, (**b**) 800%; cyclic compression loading–unloading curves of M_3_/G/PAM NDN hydrogel at strain of (**c**) 50%, (**d**) 80%.

**Figure 6 polymers-14-05247-f006:**
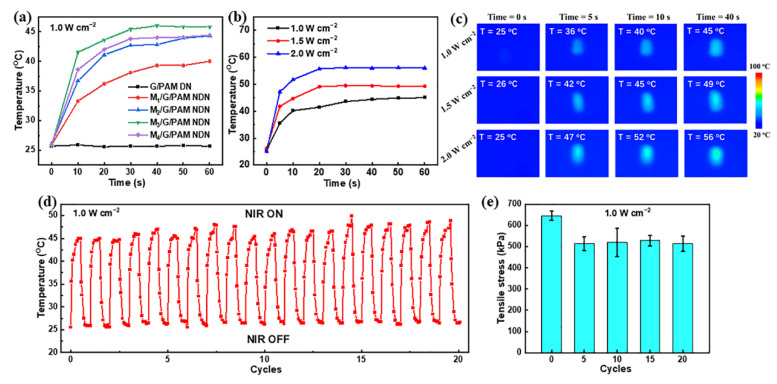
(**a**) Temperature of hydrogels with different Ti_3_C_2_T*_x_* contents upon 808 nm NIR laser irradiation (1.0 W cm^−2^); (**b**) the temperature rise curves and (**c**) real-time infrared thermal images of M_3_/G/PAM NDN hydrogel under 808 nm NIR laser irradiation with different power densities (1.0, 1.5, and 2.0 W cm^−2^); stability of M_3_/G/PAM NDN hydrogel: (**d**) 20 cycles photothermal stability under repeated laser on–off of 808 nm NIR laser irradiation (1.0 W cm^−2^); (**e**) tensile strength under repeated laser on–off of 808 nm NIR laser irradiation (1.0 W cm^−2^) (5 cycles, 10 cycles, 15 cycles, and 20 cycles).

**Figure 7 polymers-14-05247-f007:**
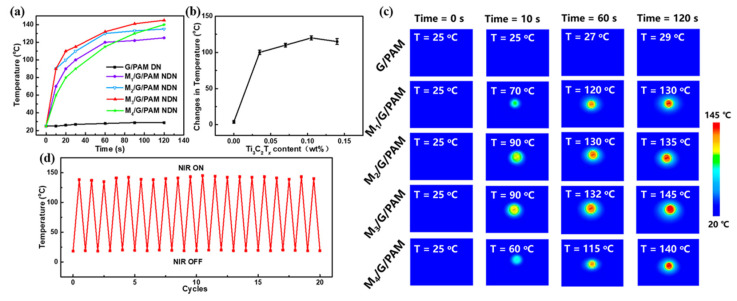
(**a**) Temperature of hydrogels with different Ti_3_C_2_T*_x_* contents upon 808 nm NIR laser irradiation (56.6 W cm^−2^); (**b**) comparison of temperature change in the hydrogels with different contents of Ti_3_C_2_T*_x_* after 120 s NIR laser irradiation (56.6 W cm^−2^); (**c**) the real-time infrared thermal images of different hydrogels; stability of M_3_/G/PAM hydrogel: (**d**) 20 cycles photothermal stability under repeated laser on–off of 808 nm NIR laser irradiation (56.6 W cm^−2^).

**Table 1 polymers-14-05247-t001:** The proportioning parameters of G SN hydrogel, PAM SN hydrogel, G/PAM DN hydrogel, and M/G/PAM NDN hydrogels.

Hydrogel	ωGelatin (wt%)	ωAM (wt%)	ωMXene (wt%)
G SN	10	0	0
PAM SN	0	28.4	0
G/PAM DN	10	28.4	0
M_1_/G/PAM NDN	10	28.4	0.035
M_2_/G/PAM NDN	10	28.4	0.070
M_3_/G/PAM NDN	10	28.4	0.105
M_4_/G/PAM NDN	10	28.4	0.140

**Table 2 polymers-14-05247-t002:** The mechanical properties of G/PAM DN hydrogel and M_3_/G/PAM NDN hydrogel.

Hydrogel	Tensile Strength (kPa)	Elongation at Break (%)	Young’s Modulus (kPa)	Energy Dissipation (MJ m^−3^)
G/PAM	205 ± 12	1172 ± 102	22 ± 3	1.24 ± 0.15
M_3_/G/PAM	646 ± 22	1105 ± 185	72 ± 4	4.45 ± 0.10

**Table 3 polymers-14-05247-t003:** Comparison of the mechanical properties of MXene-based nanocomposite hydrogels.

Hydrogel	Synthetic Method	Ti_3_C_2_T*_x_* Content (wt%)	Tensile Strength (kPa)	Compression Strength (MPa)	Reference
P(AM-AA)/CS/MXene	Initiator-induced polymerization	0.2	120	2.94 ^a^	[20]
PAA/PAM/TA/Mxene	Initiator-induced polymerization	0.5	251	—	[33]
PAM/ Gelatin/MXene	Initiator-induced polymerization	~1.894	366	0.64 ^b^	[34]
PAM/CS/ MXene	Initiator-induced polymerization	1	~75	—	[35]
PVA/MXene	Directional freezing technique	~0.4	460	—	[36]
M_3_/G/PAM	Heating–cooling and γ-ray radiation	0.105	646	3.44 ^c^	This work

^a^ Compression strength at a compression ratio of 90%. ^b^ Compression strength at a compression ratio of 80%. ^c^ Compression strength at a compression ratio of 90%.

## Data Availability

Data will be made available on request.

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
