# Peer review of "MXene/Gelatin/Polyacrylamide Nanocomposite Double Network Hydrogel with Improved Mechanical and Photothermal Properties"

_polymers, 2022, doi:10.3390/polym14235247_

Round 1
Reviewer 1 Report
The authors developed a smart hydrogel with improved mechanical properties and photothermal conversion after introducing MXene particles into the hydrogel structure. The work is well designed, and the results reached the proposed objectives. For that reason, I recommend publishing this work in its present form.
Reviewer 2 Report
Summary: This article has described the synthesis of a new type of smart hydrogel with excellent mechanical properties and photothermal conversion capability. The nanocomposite double network (NDN) hydrogel is consisted of MXene/gelatin/polyacrylamide (M/G/PAM) which has been synthesized by γ-ray radiation technology. The effect of MXene on the mechanical and photothermal effect of the NDN hydrogel has been studied.
Comments:
1. Undoubtedly, MXene incorporated hydrogels have several domains of applications. This work by Zhang et al. is well constructed as well as characterized. Before accepting in the "Polymers" journal, Authors should particularly explain the "novelty" of this work as they have mentioned in the "Abstract" section that the synthesized hydrogel is "novel".
2. Authors should mention the percentage swelling of the nanocomposite hydrogels at which they have studied the mechanical and photothermal effects. As per my understanding, at or near 100% swelling state of the hydrogel, this behaves like a brittle plastic and unable to retain its elastic nature. Author should also include the comparative study of mechanical and photothermal effect of the hydrogels at different swelling ratios.
I am recommending this manuscript to be published in “Polymers” after a minor revision.
Reviewer 3 Report
Journal: Polymers
Title: MXene/gelatin/polyacrylamide nanocomposite double network hydrogel with improved mechanical and photothermal properties
This manuscript presents the γ-radiation processable nanocomposite based dual network hydrogel systems and its applicability in photothermal applications. The results are satisfactory, and the research methodology is acceptable. I consider the present manuscript for the major revision.
Following have to be addressed,
1) I believe the photothermal performances are rapid and the temperature attained are higher enough to demonstrate the water purification experiment. I suggest authors to add some real time water purification data.
2) No absorption spectra present in the manuscript and how do authors state that upon excess filler addition, the absorption value degrades.
3) Authors should define the photothermal conversion efficiency and highlight the values achieved. Moreover, I recommend authors to add literature comparison table to prove the novel results achieved with photothermal conversion efficiency.
4) What would be the structural stability of the as-prepared hydrogels upon higher temperatures? It is rational to think that water solvent expels and collapse the entire gel matrix under sunlight conditions.
5) Figure 6e shows the deviation in tensile stress after light exposure, why such mechanical strength reduction occurs in the hydrogel systems.
6) Why the authors specifically chose NIR light laser? And what is the distance between the light source and hydrogels. Is there any difference in photothermal performances with respect to distance?
7) Using the γ-radiation process is interesting, but it has harmful effects to the human health. Authors can state their insights on γ-radiation process.
8) Is γ-radiation process suitable for the scalable production? Why don’t authors produce larger samples and what sort of complications usually occurs in forming nanocomposite based dual networks. Furthermore, how it is suitable for the industrialization?
9) Why don’t authors extend the study and connect the porosity and surface area influences in the photothermal conversion efficiency?
10) Irrespective of filler improved concentration, the absorption being lowered. Why it is so, authors can suggest some mechanism along with scheme. In M4/G/PAM system, the photothermal behavior being degraded due to the agglomeration. Authors can connect other possible factors for this degradation behavior with scientific discussions.
11) For the energy dissipation ability, how the authors calculated the energy dissipation taking tensile or compression data. If either of them, why it is so?
12) In DTG graph, M3/G/PAM hydrogel possess peak within 160-170℃, whereas other materials does not contain the peak in that range.
13) Although the MXene loading is very less, the TGA thermal decomposition curve reveals that M3/G/PAM hydrogel have residual mass around 21%. Authors have stated such thermal stability is ascribed to the hydrogen bonds. Authors can double-check the TGA results and state the clear scientific reason for such anomalous behavior.
14) Figure 2e represents the energy dissipation values of various hydrogels. Authors have to verify the energy dissipation unit (MJ/m3) and how do authors calculate the energy dissipation values.
15) Figure 2a metrics are hardly visible to see the variations. Authors can adjust the scale to get the better visibility. Figure 2d, and 2e labels have overlapped with respective y-axis.
16) Why the photothermal stability is limited to 20 cycles? What factors are governing the durability of the photothermal materials? How it is addressed in the past and how authors have used their technique to address this issue in the present project?
Round 2
Reviewer 3 Report
Journal: Polymers
Title: MXene/gelatin/polyacrylamide nanocomposite double network hydrogel with improved mechanical and photothermal properties
This manuscript presents the γ-radiation processable nanocomposite based dual network hydrogel systems and its applicability in photothermal applications. The results are satisfactory, and the research methodology is acceptable. I consider the present manuscript for publication after addressing the minor revision comments.
Following have to be addressed,
1) To maintain the mechanical stability of the hydrogel, silicone oil encapsulation was done. In real time cases, if we employ this smart hydrogels in aquatic environments, silicon oil will be released in to the drinking water resources, which will lead to serious impacts on the aquatic organisms and human beings.
2) On the other hand, the water purification by these smart hydrogels would not be efficient, if we employ this kind of encapsulation. What will be the solution for balancing this mechanical integrity and photothermal water purification?
3) In TGA analysis, 79% weight loss rate happens due to the MXene heat dissipation. If so, MXene at 400oC could retain the hydrogel matrix stable without being thermally degraded. The TGA heating process is a stable process, where the heat is supplied stepwise, how it would be possible to retain 21% of the MXene + Hydrogel weight composition at such elevated temperature.
